# Self-supervised Synthetic Pretraining for Inference of Stellar Mass Embedded in Dense Gas

## Abstract

Stellar mass is a fundamental quantity that determines the properties and evolution of stars. However, estimating stellar masses in star-forming regions is challenging because young stars are obscured by dense gas and the regions are highly inhomogeneous, making spherical dynamical estimates unreliable. Supervised machine learning could link such complex structures to stellar mass, but it requires large, high-quality labeled datasets from high-resolution magneto-hydrodynamical (MHD) simulations, which are computationally expensive. We address this by pretraining a vision transformer on one million synthetic fractal images using the self-supervised framework DINOv2, and then applying the frozen model to limited high-resolution MHD simulations. Our results demonstrate that synthetic pretraining improves frozen-feature stellar mass predictions, with the pretrained model performing slightly better than a supervised model trained on the same limited simulations. Principal component analysis of the extracted features further reveals semantically meaningful structures, suggesting that the model enables unsupervised segmentation of star-forming regions without the need for labeled data or lightweight fine-tuning.

## 1 Introduction

Stellar mass is a fundamental stellar property that governs luminosity, lifetime, stellar evolutionary tracks, and stellar nucleosynthesis, which produce the chemical elements essential for the origin of our solar system and life. In astronomy, the initial mass function (IMF) describes the stellar mass distribution at the time of their formation, that is, how many stars of a given mass are born in a star forming region (Salpeter, 1955). In general, the IMF has a downward slope, meaning that low-mass stars dominate in number; however, the small number of high-mass stars plays a crucial role by ionizing the surrounding gas, dispersing heavy elements, and providing mechanical feedback to the interstellar medium. As a result, the slope of the IMF affects not only galaxy evolution and the history of star formation, but also the origin of life, through its influence on the production of heavy elements. For example, a "shallow" slope IMF—richer in massive stars—leads to brighter young galaxies and faster chemical evolution, whereas a "steep" IMF, with fewer massive stars, produces the opposite trend.

Observations suggest that the IMF exhibits a remarkably similar shape across diverse environments, while the physical mechanisms governing this distribution remain unclear (Offner et al., 2014). To uncover the origin of the IMF, it is essential to determine the masses of young, still-forming stars (protostars and pre-main-sequence stars) through observations. However, determining the masses of young stars from direct observations remains highly challenging. They are deeply embedded within their natal molecular clouds, making them invisible with optical light. Furthermore, their luminosity originates primarily from gas accretion rather than stellar radiation, making mass estimates difficult using methods commonly applied to main-sequence stars.

Predicting stellar masses from their environments is a challenging task. The gas is highly inhomogeneous, making analytic models unreliable, while capturing the physics requires three-dimensional (3D) simulations, which are too expensive to produce in large numbers (Pelkonen et al., 2021). A promising approach is to combine high-resolution simulations with deep learning. In our work,

three-dimensional (3D) magneto-hydrodynamical (MHD) simulations are employed to capture the physics of star formation and track stellar mass growth (Nozaki et al., 2025). We then leverage two-dimensional (2D) maps of gas projected from these 3D high-resolution MHD simulations to develop deep learning models for predicting stellar masses. Since large amounts of high-resolution simulations or observational data with labels are rarely available, we propose a framework that combines self-supervised pretraining and downstream tasks. Models are trained on numerous synthetic images to learn robust visual representations, while limited high-resolution MHD simulations are reserved for evaluation on downstream tasks.

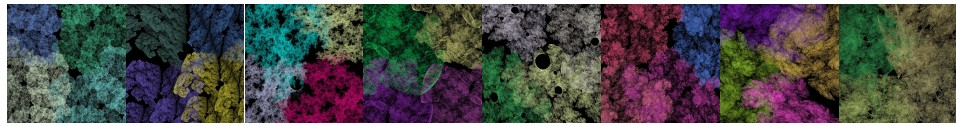

Figure 1: Examples of fractal images.

## 2 RELATED WORK

**Self-supervised Learning for Astrophysics**  Self-supervised learning has become a powerful approach for extracting image representations without labels, which could help mitigate challenges associated with limited labeled data. Early methods like MoCoV3 (Chen et al., 2021) and MAE (He et al., 2022) improved vision transformer (ViT) pretraining but often required supervised fine-tuning. In contrast, DINOv2 (Caron et al., 2021; Zhou et al., 2022; Oquab et al., 2024) captures semantic structures by enforcing consistency across multiple views, transfers to downstream tasks without fine-tuning backbones, and requires only lightweight classifiers or $k$-nearest neighbor evaluation. In astrophysics, self-supervised learning has enabled galaxy classification with sparse labels (Hayat et al., 2021; Desmons et al., 2024), inference of galaxy properties by combining simulations and observations (Eisert et al., 2024), and mitigation of observational biases through metadata (Rizhko & Bloom, 2025), with DINOv2 recently applied to galaxy images (Parker et al., 2024).

**Pretraining with Synthetic Data**  Supervised deep learning has achieved remarkable success by training models on large labeled datasets. An alternative line of research investigates the use of synthetic data generated with mathematical equations to achieve competitive performance across various downstream tasks. Such data can be produced inexpensively and in large quantities, without demanding experiments, observations, or extensive computational resources, and without raising ethical or privacy concerns. A pioneering study (Kataoka et al., 2020) showed that supervised pretraining on fractal images alone can reach competitive accuracy on natural images, in some cases even surpassing models pretrained on `ImageNet-22k` (Kataoka et al., 2022). This approach has since been extended to supervised learning with ViTs (Nakashima et al., 2022; Kataoka et al., 2022; Nakamura et al., 2023; 2024) and to self-supervised learning with convolutional neural networks (Baradad Jurjo et al., 2021; Baradad et al., 2022).

## 3 METHODOLOGY

### 3.1 DATA GENERATION

**Synthetic Images for Pretraining**  We extend the Flame algorithm (Draves & Reckase, 2008) to generate our datasets of fractal images. With randomly sampled parameters $\theta_i = (a_i, b_i, c_i, d_i, e_i, f_i)$ for rotation and shifting fed to a translation $w$, coordinates are sampled through an iterated function system (IFS; Barnsley, 1988),

$$w(\boldsymbol{x}; \theta_i) = \begin{pmatrix} a_i & b_i \\ c_i & d_i \end{pmatrix} \boldsymbol{x} + \begin{pmatrix} e_i \\ f_i \end{pmatrix}, \tag{1}$$

where $\boldsymbol{x}$ is a coordinate. At each sampling step, one of the non-linear variations (e.g., spherical and bubble) of the original Flame algorithm (Draves & Reckase, 2008) is probabilistically applied, yielding the next sampled point $\boldsymbol{x}_{i+1} = w(\boldsymbol{x}_i; \theta_i)$. Each image uses four such variations, with points from each variation rendered in a distinct color. The sampled points are then rendered as $336 \times 336$ images.

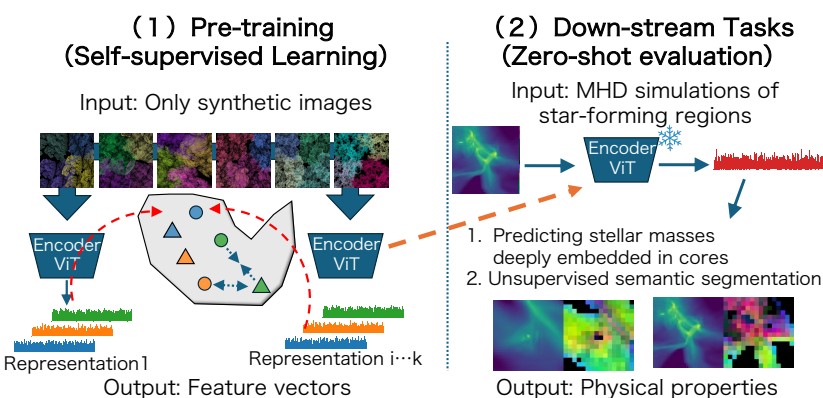

Figure 2: Overview of our model. *Left*: self-supervised pretraining with synthetic fractal images using DINOv2 to extract feature vectors. *Right*: frozen-feature or zero-shot evaluation on simulation with the frozen encoder, applied to stellar mass prediction ($k$-NN) and semantic segmentation (PCA-based colors).

We generate candidate images with one million points and retain only images whose coverage exceeds a lower threshold (the fraction of pixels covered by the fractal), preventing smudged or largely empty images while ensuring diversity within each image (Kataoka et al., 2020; Anderson & Farrell, 2022). In our setup, the threshold is set to 0.9, motivated by the hypothesis that higher coverage increases inter-image diversity and improves accuracy; this hypothesis is examined in Section 6.1. To speed up sampling, we estimate coverage on an eight-times downsampled image with $42 \times 42$ pixels. Examples are shown in Fig. 1. The dataset generation for 1M synthetic images was executed with a throughput of 2.67 TFLOPS per image and 2.67 EFLOPS in total.

**Simulations of star-forming regions**   We performed 3D MHD simulations with SFUMATO, an adaptive mesh refinement code (Matsumoto, 2007; Matsumoto et al., 2015; Fukushima & Yajima, 2021; Nozaki et al., 2025), in a cubic box of 4 parsec[1] per side containing 3000 $M_\odot$[2] of gas with an initial uniform proton density of 1365 cm$^{-3}$ and a magnetic field of 10 $\mu$G along the $z$-axis. We solve the following basic MHD equations with the Poisson equation:

$$\frac{\partial \rho}{\partial t} + \nabla \cdot (\rho \mathbf{v}) = 0, \tag{2}$$

$$\frac{\partial}{\partial t}(\rho \mathbf{v}) + \rho(\mathbf{v} \cdot \nabla)\mathbf{v} = -\nabla P + \frac{1}{4\pi}(\nabla \times \mathbf{B}) \times \mathbf{B} - \rho \nabla \Phi, \tag{3}$$

$$\frac{\partial}{\partial t}(\rho E) + \nabla \cdot \left[ \left( \rho E + P + \frac{|\mathbf{B}^2|}{8\pi} \right) \mathbf{v} - \frac{(\mathbf{v} \cdot \mathbf{B})}{4\pi}\mathbf{B} \right] = -\rho \mathbf{v} \cdot \nabla \Phi + \Gamma + \Lambda, \tag{4}$$

$$\frac{\partial \mathbf{B}}{\partial t} = \nabla \times (\mathbf{v} \times \mathbf{B}), \tag{5}$$

$$\nabla^2 \Phi = 4\pi G \rho, \tag{6}$$

$$E = \frac{|\mathbf{v}|^2}{2} + (\gamma - 1)^{-1}\frac{P}{\rho} + \frac{|\mathbf{B}^2|}{4\pi\rho}, \tag{7}$$

where $\rho, P, \mathbf{v}, \boldsymbol{B}, \Phi, E, \Gamma$ and $\Lambda$ are the density, pressure, velocity, magnetic field, gravitational potential, total energy, the heating and cooling rates. The heating $\Gamma$ and cooling $\Lambda$ rates include the processes such as heating from chemical reactions, cooling from line emissions and energy transfer between gas and dust.

To follow the long-term evolution, we use the sink particle method (Matsumoto et al., 2015), in which unstable dense clumps are replaced by sink particles that accrete gas within a fixed radius

---

[1]1 parsec $\sim 3.08 \times 10^{13}$ km $\sim 3.26$ light years

[2]1 $M_\odot$ (solar mass) is equal to the mass of the Sun.

$(5.0 \times 10^{-4}$ pc). The accreted mass is taken as the protostellar mass, allowing us to trace protostellar growth. The finest spatial resolution is $\Delta x \sim 3 \times 10^{-3}$ parsec, sufficient to resolve the Jeans length with more than five cells, given an initial velocity field with a Mach number of 10. Our dataset consists of 32k snapshots of 0.5 pc regions centered on protostars, from which we construct $64 \times 64$ maps of column density, mean line-of-sight velocity, and its velocity dispersion along the $x$, $y$, and $z$ axes, each paired with the elapsed time since protostar formation and the stellar mass shown in Fig. 2 (2). The simulation was executed with a throughput of 2540 TFLOPS per snapshot and 81.2 EFLOPS in total.

### 3.2 Model Implementation

We employ a ViT-L/16 encoder within the DINOv2 framework (Caron et al., 2021; Zhou et al., 2022; Oquab et al., 2024) for self-supervised pretraining and zero-shot or reasonable fine-tuning evaluation. The encoder is pretrained on 1M synthetic fractal images at a resolution of 336 for 100 epochs with a batch size of 1024 and a patch size of 16. For comparison, we implement a ResNet-18 (He et al., 2016) baseline trained in a fully supervised manner. Both models use a cosine-annealed learning rate schedule with a maximum of 0.04, including 10 warm-up epochs followed by 90 epochs of cosine decay. The ResNet-18 is trained with an $L_2$ regression loss and a batch size of 1024 for 100 epochs. Simulation data are preprocessed by applying a logarithmic transformation to stellar mass and column density, and min–max normalization to mean line-of-sight velocity and its dispersion. To assess scaling, we further pretrain a ViT-L/16 encoder on 10M synthetic fractal images with a batch size of 2048 for 70 total epochs (10 warm-up epochs followed by 60 cosine-decay epochs), due to computational limitations.

## 4 Experimental Setup

**Self-supervised Pretraining with Synthetic Data and $k$-NN Regression**  The pretrained ViT-L/16 encoder is applied to 32k snapshots from star-formation simulations at a resolution of 64 to obtain 1024-dimensional feature vectors. Principal component analysis (PCA) is fitted on the training split and applied to all features while preserving the full dimensionality of 1024 (PCA whitening). The transformed features are then evaluated with a distance-weighted $k$-nearest neighbors ($k$-NN) regressor ($k = 5$) to predict the logarithm of stellar mass, using 24k training and 8k test samples. Prediction performance is assessed in terms of root-mean-square error (RMSE) and the coefficient of determination ($R^2$).

**Zero-shot Unsupervised Feature Visualization**  To examine the semantic structure of the learned representations, feature vectors from the pretrained ViT-L/16 encoder are projected with PCA and the first three components are mapped to the RGB color space (dimensionality reduction on PCA). Although pretraining is performed with a patch size of 16, for visualization we linearly upsample $4\times4$ input patches to $16\times16$ prior to encoding, in order to maintain consistency with the token granularity of the model.

## 5 Results

We evaluate the ViT-L/16 encoder, pretrained on fractal images, on two downstream tasks, keeping the model parameters frozen.

### 5.1 Frozen-feature Regression on Stellar Masses

Fig. 3a shows a scatter plot of the first and second PCA components of feature vectors from column density, mean line-of-sight velocity, and its velocity dispersion maps. The distribution exhibits a weak trend with stellar masses, which are otherwise difficult to infer from 2D density and velocity information alone.

To evaluate predictive performance, we use all all PCA-whitened components with a $k$-NN regressor trained on the training set and tested on the validation set. Fig. 3b and Fig. 3c compare stellar mass predictions from our method and a supervised ResNet-18 baseline, respectively. Both approaches

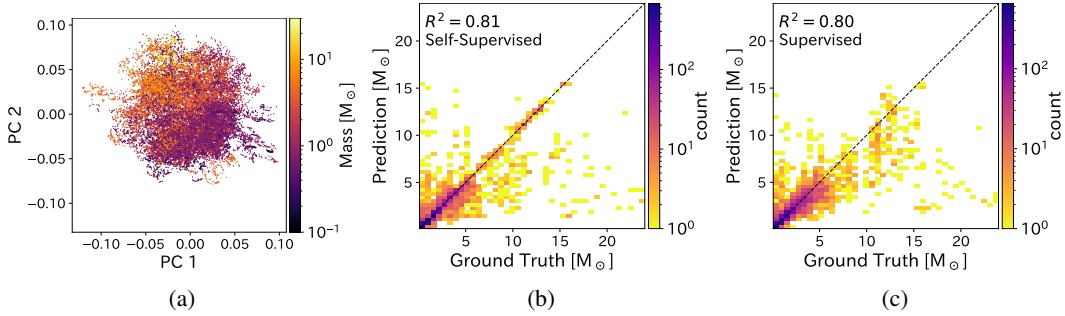

Figure 3: Frozen-feature regression of stellar masses. **(a)** PCA projection of feature vectors from DINOv2 colored by stellar mass. **(b)** True versus predicted stellar masses using DINOv2 representations with $k$-NN regression. **(c)** True versus predicted stellar masses from a supervised ResNet-18 baseline.

follow the ground truth trend up to $\sim 6\,\mathrm{M}_\odot$, where more than $10^2$ training samples are available. Beyond this, with fewer than 10 samples, ResNet-18 tends to underestimate stellar masses, while DINOv2 captures many true values in the range $6$–$15\,\mathrm{M}_\odot$. At higher masses, neither model performs reliably due to data scarcity. Table 1 shows that PCA features slightly improve scores over raw feature vectors and that pretraining with fractal images markedly improves performance compared to the result of random initialization with DINOv2.

| Methods | ResNet-18 | | DINOv2 + $k$-NN ($k = 5$) | | |
| | Random Init. | Pretrained | Random Init. | Pretrained | with PCA whitening |
|---|---|---|---|---|---|
| $R^2(\uparrow)$ | -1.9 | 0.80 | -0.58 | 0.80 | **0.81** |
| RMSE ($\downarrow$) | 0.34 | 0.089 | 0.52 | 0.089 | **0.088** |

Table 1: $R^2$ and RMSE of frozen-feature regression on stellar mass using ResNet-18 and DINOv2.

## 5.2 ZERO-SHOT SEMANTIC SEGMENTATION WITH PCA-BASED COLORS

Fig. 4 shows four examples (a–d), each containing four panels: column density $N_{\mathrm{HI}}$, mean line-of-sight velocity $v_{\mathrm{los}}$, its velocity dispersion $\sigma_v$, and a color map based on the first three PCA components of the 1024-dimensional feature vectors. Black areas correspond to either diffuse, low-density regions or regions of very high velocity dispersion, the latter likely marking sites of ongoing star formation. Yellow to yellow-green areas highlight regions of low velocity dispersion (Fig. 4a–4c). Magenta and dodgerblue indicate negative and positive line-of-sight velocities in regions of high velocity dispersion (Fig. 4b–4d), where gas may accrete onto dense cores and contribute to stellar growth. Notably, this semantic segmentation arises directly from the PCA projection of pretrained representations, without any labeled data or supervised fine-tuning.

## 6 ABLATION STUDY

### 6.1 COVERAGE ON SYNTHETIC PRETRAINING DATA AND ACCURACY ON DOWNSTREAM TASKS

We construct the pretraining dataset by applying a lower coverage threshold and retaining only images that exceed it, thereby avoiding degenerated cases from specific fractal parameters (e.g., contracted smudged shapes or largely empty images) that worsen the performance of representation learning. While the main experiments fixed the lower coverage at 0.9, here we sweep it more broadly and study its effect on downstream accuracy.

In this evaluation, we pretrain a ViT for 100 epochs on 100k synthetic images with a batch size of 256, and evaluate the accuracy using a $k$-NN classifier ($k = 10$) on frozen ViT features for `ImageNet-1k`. We vary the lower coverage threshold from 0.0 to 0.9. Table 2 shows top-1 and

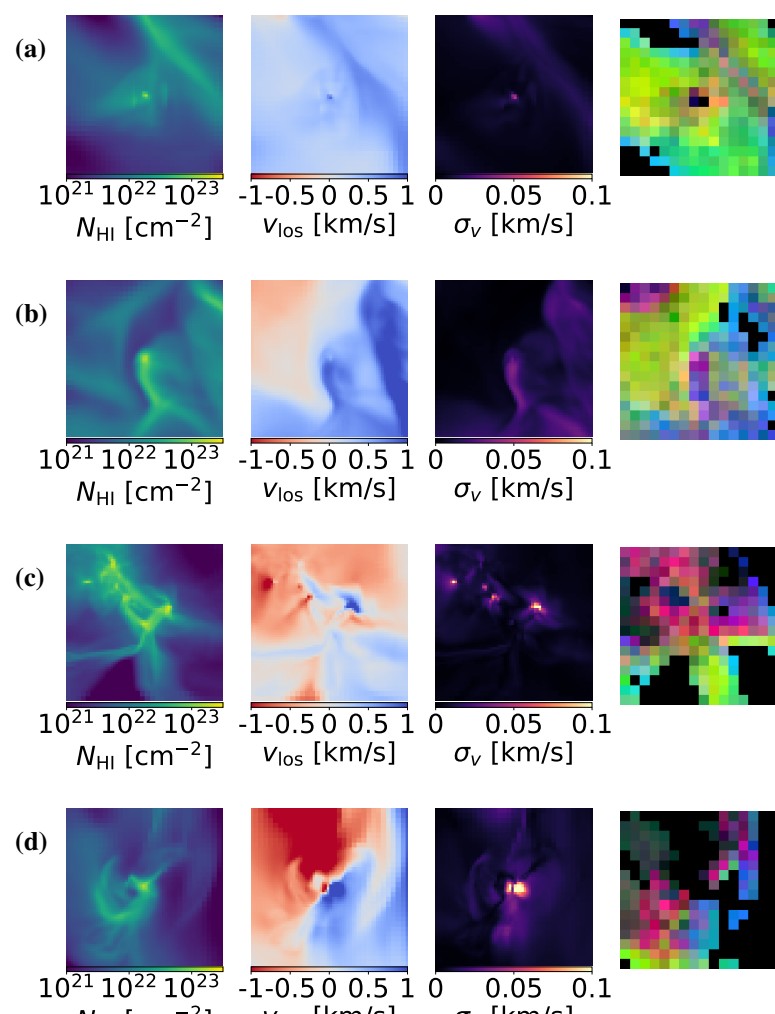

Figure 4: Snapshots from MHD simulations with visualizations of PCA components of feature vectors. Each panel shows four maps: column density $N_{HI}$, mean line-of-sight velocity $v_{los}$, its velocity dispersion $\sigma_v$, and a color map of the first three PCA components from image patches.

| | w/o PCA whitening | | w PCA whitening | |
| --- | --- | --- | --- | --- |
| Lower Coverage Threshold | top-1 | top-5 | top-1 | top-5 |
| 0.0 | 7.44 | 14.18 | 15.21 | 24.74 |
| 0.3 | 10.41 | 18.85 | 21.11 | 32.39 |
| 0.4 | 9.62 | 17.89 | 19.94 | 31.28 |
| 0.5 | **11.74** | **20.24** | **21.20** | **32.39** |
| 0.6 | 8.02 | 14.03 | 12.83 | 20.85 |
| 0.9 | 8.58 | 14.58 | 14.36 | 22.69 |

Table 2: Effect of lower coverage threshold during synthetic pretraining on downstream `ImageNet-1k` accuracy. Those are measured on top-1 and top-5 accuracy with a $k$-NN classifier ($k = 10$) on frozen ViT features, with and without PCA whitening.

top-5 accuracies, with and without PCA applied for the raw feature vectors. Although our initial hypothesis was that enforcing high coverage (threshold = 0.9) would increase inter-image diversity and thus improve accuracy, we find that a moderate threshold around 0.5 yields the best performance across all cases.

## 6.2 SCALING ON PRETRAINING DATA

To assess pretraining data scaling, we evaluate a frozen ViT with a $k$-NN classifier ($k = 10$) and with a trainable MLP head ("MLP probe") after pretraining on 1M and 10M synthetic images. In $k$-NN, labels of the valid data are predicted using the frozen features of the training data, whereas the MLP probe performs lightweight fine-tuning by training an MLP head on those feature vectors while keeping the encoder frozen. For the MLP probe, we sweep the head learning rate from 0.65 to 0.85; 0.85 performs best, although 100 epochs might be insufficient for full convergence.

Table 3 shows that the MLP probe achieves 36.8% top-1 and 58.9% top-5 accuracy on `ImageNet-1k` with pretraining on 10M synthetic images, compared to 32.6% and 53.5% with 1M, while the accuracies on $k$-NN improve only marginally. Scaling to 10M yields broadly positive but modest gains for both $k$-NN and the MLP probe, suggesting redundancy in the larger dataset and that diversity—not volume alone—drives discriminative feature learning.

| | $k$-NN on I-1k w/o PCA whitening | | $k$-NN on I-1k w/ PCA whitening | | MLP probe on I-1k | | $k$-NN on SF with PCA whitening | |
| Dataset size | top-1 | top-5 | top-1 | top-5 | top-1 | top-5 | $R^2(\uparrow)$ | RMSE ($\downarrow$) |
|---|---|---|---|---|---|---|---|---|
| 1M | 8.58 | 14.58 | 14.36 | 22.69 | 32.6 | 53.5 | 0.81 | 0.088 |
| 10M | 8.40 | 15.90 | 14.50 | 24.64 | 36.8 | 58.9 | 0.82 | 0.086 |

Table 3: Effect of pretraining dataset size on downstream performance: `ImageNet-1k` (I-1k) top-1/top-5 using $k$-NN ($k$=10) with and without PCA whitening and an MLP probe on frozen ViT features, and coefficient of determination ($R^2$) and root-mean-square error (RMSE) on 3D MHD simulations of star-forming (SF) regions. All evaluations use a frozen ViT encoder.

## 7 SUMMARY AND LMITATIONS

Our results demonstrate that self-supervised synthetic pretraining can serve as a data-efficient alternative to supervised pipelines in high-resolution, yet data-limited, MHD simulations, with a ViT encoder achieving performance comparable to that of supervised learning. PCA-based visualization further revealed meaningful structures, such as dense cores and inflows, motivating the extension of this approach to stellar property prediction and its direct application to observational data.

We also find two practical choices that matter for synthetic pretraining. First, varying the lower coverage threshold reveals that a threshold of approximately 0.5 outperforms very high coverage of 0.9, suggesting that aggressive filtering may suppress morphological diversity and harm downstream accuracy (Section 6.1). Second, scaling the pretraining set from 1M to 10M images improves the accuracy of the linear evaluation on `ImageNet-1k` by 4.2% in top-1 and 5.4% in top-5. Nevertheless, the gain is modest, indicating redundancy in the 10M dataset and suggesting that diversity—not volume alone—governs the ViT's ability to learn discriminative features (Section 6.2).

The present approach indicates a potential to predict protostellar masses, and incorporating information from more extended gas and velocity fields may ultimately enable predictions of the final stellar mass and the IMF. The framework, however, still relies on labeled simulation data for training in order to apply it to observations. Meanwhile, PCA-based segmentation highlights that broad structural patterns can be identified without labels, though addressing observational noise—potentially by constructing datasets from the noise itself—will be crucial for robustness in practice. Domain transferability among fractal images, simulations, and observations should also be investigated in future work.

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

## A    DECLARATION OF LLM USAGE

We used GPT-4 to generate codes that evaluated the transformed features using the $k$-NN (see Section 4) and that created the figures presented in the main text. GPT-4 was also used as an aid in searching for related works (Section 2). All outputs from GPT-4 were carefully reviewed and validated by the authors before inclusion in the paper.

