# OpenReview forum: "Self-supervised Synthetic Pretraining for Inference of Stellar Mass Embedded in Dense Gas"
_ICLR.cc/2026/Conference — Submitted to ICLR 2026_

### Official Review · Reviewer_KVbC · 2025-10-25

**Soundness:** 2
**Presentation:** 3
**Contribution:** 1
**Rating:** 2
**Confidence:** 4

**Summary:**

The authors probe the effectiveness of synthetic pre-training for stellar mass prediction. To do so, they create a fractal-based synthetic dataset, train DINOv2 on it, then evaluate the frozen features for the task with kNN classification, where it performs on par with a supervised ResNet18. Finally, they provide some ablations on the synthetic dataset construction and size.

**Strengths:**

The motivation to use synthetic data to overcome the limitations encountered when training on data from expensive simulations is sound and well-illustrated by the stellar mass prediction task. The paper is clearly written, and the visualizations aid in the understanding of the different parts of the work.

**Weaknesses:**

While the paper is well-motivated, the experimental setup and results do not show that the proposed contributions are indeed the best choice. The fact that a simple ResNet18, which is far smaller than DINOv2, performs almost identically to the final model with synthetic data defeats the purpose of the proposed pipeline.  Moreover, a crucial experiment that is missing is comparing the DINOv2 weights obtained on the synthetic dataset to simple ImageNet (or other) pre-trained weights. This would show that the fractals are indeed a better pre-training dataset than the standard model weights.

**Questions:**

- The scaling results should be done with the same settings to isolate the effect of dataset size. For instance, they should use the same batch size.
- Table 2 shows that the settings with which the synthetic dataset was made, which were used for the main experiments, are far inferior to using a coverage threshold of 0.5. How would the main results change with the optimal settings?
- The chosen values for the LR sweep (from 0.65 to 0.85) are rather uncommon. Why were these values used?
- Why did the authors not follow the common practice of using an MLP on the frozen DINOv2 features, especially as it performs far better than kNN on Table 3?
- The description on L189-190 should be improved: "Principal component analysis (PCA) is fitted on the training split and applied to all features while preserving the full dimensionality of 1024 (PCA whitening)." Applying PCA while preserving the full dimensionality is not the same as whitening. As kNN results change after the operation, there is a rescaling of the principal components involved.
- The method section would benefit from some more information on DINOv2 and how it is trained, especially as there is plenty of space left.

---

### Official Review · Reviewer_EfmV · 2025-10-28

**Soundness:** 2
**Presentation:** 3
**Contribution:** 2
**Rating:** 4
**Confidence:** 4

**Summary:**

This paper explores the application of self-supervised visual representation learning, specifically DINOv2, to magneto-hydrodynamic (MHD) simulations in an astrophysical context. The downstream tasks include stellar mass regression and zero-shot semantic segmentation, with visualization based on PCA color mapping of learned features.

**Strengths:**

- Applying machine learning methods to astrophysical simulation data is a promising and relatively under-explored direction.

- The paper is clearly written and easy to follow, making it accessible to both ML and astrophysics audiences.

**Weaknesses:**

- Limited technical novelty: The work represents a direct application of DINOv2 to MHD simulation data, without introducing new algorithmic insights, model architectures, or training paradigms. The methodological contribution is therefore somewhat minimal from an ML perspective.


- Outdated baselines: The experimental comparisons are limited to ResNet-18, omitting several recent and more advanced methods tailored to similar physical simulation contexts. For instance, diffusion-based models such as [a] would be natural points of comparison.


- Unclear motivation and empirical gain: The paper lacks a strong motivation for why DINOv2 in particular is well-suited to this application. Additionally, the empirical gains over baselines appear marginal, and the choice of metrics and evaluation protocols could be better justified.


- Missing discussion on computational cost: There is no mention of the computational resources or training/inference time required by pre-training. This information would be especially relevant for the astrophysics community, where large-scale computing is often a limiting factor in adoption.

While the topic is interesting and the results may be of value to the astrophysics community, it may be better suited for publication in astrophysical venues such as The Astrophysical Journal, rather than ML venues like ICLR.

---
[a] Denoising Diffusion Probabilistic Models to Predict the Density of Molecular Clouds, APJ 2023.

**Questions:**

How sensitive is the performance in terms of the ML hyper-parameters such as the map resolutions? Currently, the paper does not include discussions on any ML design choices.

---

### Official Review · Reviewer_NH2M · 2025-10-30

**Soundness:** 2
**Presentation:** 3
**Contribution:** 3
**Rating:** 2
**Confidence:** 3

**Summary:**

The paper’s method is a proof-of-concept pipeline showing that self-supervised pretraining on synthetic fractal images can yield transferable visual features for astrophysical inference. A Vision Transformer (ViT-L/16) is first pretrained with the DINOv2 self-supervised objective, learning to produce consistent feature representations across multiple augmented views of millions of generated fractal textures without labels. This pretrained encoder is then frozen and applied to 2D projection maps from 3D magnetohydrodynamic simulations. The resulting 1024-D embeddings are preprocessed with PCA and fed into a k-NN to predict each snapshot’s log protostellar mass. Additionally, principal-component maps of the frozen features are visualized as RGB “zero-shot segmentations,” revealing structural patterns without supervision. The overall purpose is to test whether synthetic self-supervised pretraining can provide useful representations that outperform or complement fully supervised models trained directly on limited simulation data.

**Strengths:**

- The paper is well motivated and aligns well with current trends toward developing low-cost, easily deployable methods.

- The idea of training on easily generated fractals is both novel and clever. I find it genuinely interesting and promising for this application.

- The experimental pipeline is clearly described and relies on well-established techniques such as PCA and kNN.

- The proposed method demonstrates improvement over a simple supervised baseline.

**Weaknesses:**

1. The main weakness I find is the lack of a clear baseline that aligns with the paper’s “limited data” motivation. The supervised ResNet-18 trained from scratch seems to me like a rather naive baseline. I understand that the goal was to demonstrate that synthetic-pretrained, frozen ViT features generalize better than a fully supervised model trained with limited data. However, the presented experiment does not seem to reflect a truly limited-data setting. The 24k–8k split they use already falls into a relatively large range where a fully supervised model might start performing well, perhaps explaining the small performance gain reported (0.81 vs. 0.80). The paper would benefit from stronger baseline selection (Is ResNET-18 truly the "best" supervised baseline?) and experiments under genuinely limited-data regimes. For example, exploring smaller training sizes (e.g., 100, 1,000, 10,000 samples) would better illustrate how well their method performs with scarce data, which I believe is the main goal of the paper. I suspect the authors’ approach could indeed outperform fully supervised methods in such regimes, but since this is not shown in the paper, that remains only a conjecture.

2. It seems to me that the 30 K-sample dataset used here serves as a simplified, single-setup demonstration. This may be sufficient for testing the concept of fractal pretraining, but it falls short of the frontier of astrophysical simulation realism. If the reported results were stronger (e.g., more than the marginal 0.81 vs 0.80 improvement), the argument would be more compelling. As it stands, I wonder whether the observed gain is simply due to the simplified simulation setup. Do the authors have access to higher-fidelity simulated data that could better demonstrate robustness? I understand that limited simulation realism might be part of the motivation for the proposed methodology, but some additional evidence is needed to convincingly support the claim.

3. I do not see any experiments involving real data (only simulations). I suggest that the authors consider testing the method on a small sample of real telescope images (or any other relevant observational dataset) to corroborate the results. I realize this may be challenging, but without such validation, the work reads more like a controlled experiment than a method ready for practical use.

4. The pretraining ablations are informative, but the coverage-threshold and scale analyses mainly reference ImageNet. A concise ablation directly on the astrophysics task, such as varying the amount of synthetic data (e.g., 1 M vs 10 M samples) or adjusting the fractal coverage thresholds, would make the study more complete and self-contained.

Editorial suggestion:

Even though I understand the motivation of the method, and I believe is very interesting; I think the introduction doesn't clearly articulate it, and might confuse the standard ICLR reader. For example, a more logical flow (inspired in your current intro) would be:
1. Why IMF is relevant
2. Masses can’t be measured directly; simulations are expensive
3. Supervised ML needs many labeled simulations, which are costly and limited in astrophysics
4. Idea: Use self-supervised synthetic pretraining on cheap fractal images to learn general structure representations.
5. Brief explanation of the method
6. Summary of contributions and results

Minor comment:

- Typo in SUMMARY AND LMITATIONS

**Questions:**

Most of my questions are already part of the weaknesses above, but here I try to summarize:|

- Why was a 24k/8k split chosen to represent a “limited data” regime, and how would the method perform with smaller subsets (e.g., 1k–10k samples)?

- Is ResNet-18 the most appropriate supervised baseline, or could a stronger model better contextualize the claimed improvements? Did you try other architectures?

- Can the authors validate their method on higher-fidelity simulations or real telescope data to demonstrate robustness beyond simplified setups? Can you illustrate on how much do you actually gain by using your method over more robust alternatives? (e.g computational and/or financial cost)

- How sensitive is the approach to the amount and structure of the synthetic (fractal) data used for pretraining? Specially under this specific astrophysics application?

- How do you envision this method to be used in practice? Would you pretrain once and for all, e.g generate a lot of fractal images, and pre-train, and the keep those frozen always and then do step (2)? or would you do the pre-training every time depending on the specific type of images available for the relevant study?

---

### Official Review · Reviewer_FNQu · 2025-10-31

**Soundness:** 1
**Presentation:** 2
**Contribution:** 1
**Rating:** 2
**Confidence:** 4

**Summary:**

The paper introduces a novel approach for estimating stellar mass by leveraging self-supervised pretraining. Specifically, the authors first pretrain a DinoV2 model on fractal images, and subsequently apply this pretrained model to images generated from magnetohydrodynamic (MHD) simulations for the task of stellar mass prediction. The proposed method is compared against a fully supervised baseline using a ResNet architecture, and is shown to achieve superior performance. Additionally, the paper presents visualizations of principal component analysis (PCA) feature maps to provide further insight into the learned representations.

**Strengths:**

- The description of the problem provides a compelling justification for adopting a self-supervised approach, effectively highlighting the limitations of traditional supervised methods in this context.

- The figures are well-designed and informative, contributing to the clarity of the presentation.

**Weaknesses:**

- **Lack of Justification for Pretraining Strategy:** The primary shortcoming of the paper is that it does not adequately address why the original pretrained DinoV2 features cannot be used directly. The necessity of pretraining the entire model is not clearly justified, especially given that pretraining typically requires substantial computational resources and large datasets, neither of which are discussed in detail.

- **Unconvincing Results:** The results presented in Table 1 are not compelling, as the reported performance would be comparable to the baseline if error bars where accounted for.

- **Lack of Comparison to Existing Methods:** The approach is not compared against any established methods for stellar mass estimation, making it difficult to assess its relative merit.

- **Insufficient Presentation Length:** The paper is notably brief (7 pages) and does not meet the standard length or depth of presentation typically expected at ICLR.

**Questions:**

- Why not use the original DinoV2 features for this task ?

---

> ### Author Response · Authors · 2025-11-19
>
> We would like to thank the reviewer for the detailed review and insightful feedback.
>
> We did not use the original DINOv2 features because we expected that applying features pretrained with natural images to scale-free structures in astrophysical data could degrade performance due to the domain shift (e.g., Farahani et al. 2020).
>
> Star-forming regions in the Universe—i.e., turbulent molecular clouds—are known to exhibit self-similar, scale-free density structures (e.g., Elmegreen et al. 2004). In contrast, natural images contain multiple objects and semantic elements and generally lack self-similarity, which could lead to a substantial domain shift.
> For this reason, we used fractal images—a simple example of scale-free, self-similar structures—as the pretraining data in our approach.
> As an additional experiment, we also evaluated a ViT-L/14 model using the official LVD-142M pretrained parameters (trained on 142M natural images) and confirmed the performance was slightly worse than that of our method.
>
> |              |   LVD-142M | Ours (Fractals) |
> |---------------------------|---------|--------------|
> | $R^2 (\uparrow)$      | 0.78        | 0.81 |
> |RMSE $(\downarrow)$      | 0.089            | 0.088    |

---

### Official Review · Reviewer_PJRL · 2025-10-31

**Soundness:** 3
**Presentation:** 3
**Contribution:** 1
**Rating:** 2
**Confidence:** 4

**Summary:**

This paper proposes a step towards the answering the question of estimating stellar masses from astronomical observations. To do so, the paper argues that magnetohydrodynamical (MHD) simulations are essential to link theory to observations; however, those simulations are too expensive to allow using them as the forward model to analyze data, motivating the need for an emulator. The paper proposes a method to train such an emulator in a situation where labeled data is scarce. The authors propose a self-supervised synthetic pretraining (SSP) approach. Specifically, they pretrain a Vision Transformer (ViT) encoder using the DINOv2 framework on a large dataset of one million synthetic fractal images. This pretrained, frozen encoder is then used as a feature extractor on the small, high-resolution MHD simulation dataset. The extracted features are used for two downstream tasks: 1) predicting stellar mass via k-NN regression and 2) unsupervised semantic segmentation via PCA visualization. The authors claim this SSP approach achieves slightly better regression performance ($R^2 = 0.81$) than a supervised ResNet-18 baseline ($R^2 = 0.80$) trained on the same limited data.

**Strengths:**

**1. This is a well-motivated problem:** This is a well-motivated and common significant problem is the field of computational astrophysics. Often, the only available link between latent parameters of interest and observations are simulation data that are too expensive to generate in large quantities. Using ML in this low-data regime is a well-justified goal.

**2. The approach is well-motivated and sensible:** The core idea of the paper, that is, to leverage large, cheap-to-generate synthetic data (fractals) for self-supervised pretraining before applying or fine-tuning on a small expensive target data set is logical. Synthetic pretraining is a potentially useful paradigm in other scientific domains.

**3. The material is well-presented:** The paper is well-written and easy to follow. The methodology, data-generation process, and experimental setup are clearly described.

**4. The authors present some methodological ablation:** The authors provide ablation studies on the impact of the fractal coverage threshold and the size of the pretraining dataset  (evaluated on both ImageNet-1k and the primary task), which offer some useful insights for this specific pretraining pipeline.

**Weaknesses:**

**1. The work lacks methodological novelty for ICLR venue:** The primary weakness is the paper's limited methodological contribution to the machine learning community. The work appears to be a direct application of an existing, off-the-shelf SSL framework (DINOv2) combined with an existing pretraining data concept (fractals). There are no apparent modifications or novel insights into the DINOv2 algorithm, the ViT architecture, or the learning process itself. The downstream tasks are handled by standard, non-novel methods (k-NN regression and PCA) .

**2. Limited scope and demonstrated generalizability of work:** The paper's claims are validated on a single, highly specific application. While the introduction of this astrophysics task is interesting, the paper does not demonstrate that the method (DINOv2 + fractal pretraining) is a generally useful tool for the broader ML community. The included ablations on ImageNet-1k are a good step but are not the focus and the performance is not strong enough to make a general claim.

**3. There is only a weak and limited baseline comparison:** The main experimental result hinges on a comparison between the proposed SSP-DINOv2-ViT (frozen feature extractor) and a supervised ResNet-18. To me this feels like an apple-to-orange comparison, because both the architecture and the training paradigm are simultaneously modified and it's not clear to which the claimed improvements are due. I would have liked to see a more comprehensive comparison that would include:

- A comparison with standard transfer learning using the ViT architecture, with the pertaining on ImageNet-22k, used as a frozen feature extractor. This would show that improvements are actually due to the fractal pretraining.

- A fully supervised equivalent architecture, using the ViT trained on the same limited MHD dataset.

- A comparison with alternative SSL methods pertained on the same 1M fractal dataset.

**4. The performance gained claimed are marginal:** The central claim of superiority rests on an R² score of 0.81 for the proposed method versus 0.80 for the supervised ResNet-18 baseline. This is a negligible improvement and does not provide a compelling argument for the adoption of this complex pretraining pipeline, especially given the weak baseline. If there is different metric according to which the improvement is more obvious or significant, it's not clearly described in the paper.

To me, it appears like the stated goal of the paper is to infer stellar masses from their environment, and to use the trained model as an emulator for MHD simulations. The paper fails to connect its results to the stated ultimate scientific objective. I would have expected the paper to evaluate and benchmark different emulators on their performance within an inference framework (e.g. SBI), to estimate stellar masses from mock data. The fact that the paper only reports $R^2$ values doesn't allow the reader to judge if the precision is sufficient for the downstream task. There is no discussion of: what is the required emulator precision to avoid biasing the resulting stellar mass posteriors, and how the proposed model or the baseline actually perform on this, and how do emulators need to be improved to meet the science requirement? There is also no discussion of OOD robustness. This is a major flaw, because it makes it very difficult to judge the actual utility of the work.

**5. Audience mismatch:** Given the lack of ML novelty and the single, highly domain-specific application, this paper seems poorly aligned with the ICLR audience. The contribution is almost entirely scientific (i.e., "a new way to estimate stellar mass") rather than methodological (i.e., "a new way to perform self-supervised learning"). This is reflected in the introduction which is just an astrophysics introduction. This work would likely be a much stronger fit for a computational astrophysics journal or a workshop focused on machine learning for the physical sciences.

**Questions:**

1. How does the fractal-pretrained model compare to a ViT pretrained on a large natural image dataset (like ImageNet-22k) when applied as a frozen feature extractor to your MHD simulation data?

2. Why was a supervised ResNet-18 chosen as the primary baseline? To make a fair claim about the benefits of the pretraining strategy, wouldn't a supervised ViT (the same architecture) be the more appropriate control experiment?

3. Could you clarify what you see as the primary machine learning contribution of this work?

4. The paper states the goal is to infer stellar masses from their environments, which scientifically implies using the model as an emulator to infer mass from observations. Could you elaborate on the connection between the reported  performance (R²=0.81) and this ultimate scientific goal? What is the required precision for an emulator in this domain to be scientifically useful within an inference framework (e.g., to not bias the posterior mass estimates), and how does your model's performance compare to this requirement?

---

> ### Author Response · Authors · 2025-12-01
>
> We thank the reviewer for the thoughtful and constructive comments.
>
> # 1 Comparison with ImageNet-pretrained ViT models
>
> We performed an additional experiment using a ViT-L/14 model distilled and pretrained on 142M natural images (LVD-142M) as a frozen feature extractor. Applied to our MHD simulation dataset, it achieves:
>
>
> |              |   LVD-142M | Ours (Fractals) |
> |---------------------------|---------|--------------|
> | $R^2 (\uparrow)$      | 0.78        | 0.81 |
> |RMSE $(\downarrow)$      | 0.089            | 0.088    |
>
> Thus, our fractal-pretrained model performs better than the model with 142 M natural images (LVD-142M). We attribute this to the mismatch between natural images and the scale-free, turbulence-driven structures of astrophysical simulations.
>
> # 2 Choice of ResNet-18 as the supervised baseline
> Our dataset contains only 22k samples, which limits the feasibility of training large-capacity architectures from scratch. For this reason, we selected ResNet-18 as a capacity-appropriate supervised baseline.
> For completeness, we also trained a ViT-L/16 directly as a supervised learning on the 22k labeled samples. As expected, the model overfits heavily and exhibits low performance:
>
> |              |   ViT-L/16 (SL) |Ours (SSL) |
> |---------------------------|---------|--------------|
> | $R^2 (\uparrow)$      | 0.21        |  0.81 |
> |RMSE $(\downarrow)$      | 0.18    | 0.088    |
>
> This indicates that a supervised ViT-L is likely not well-suited as a baseline in our data-limited setting, while self-supervised fractal pretraining seems to help ViTs act as more effective feature extractors despite the small number of downstream labels.
>
> # 3 Primary machine-learning contribution
>
> The primary machine-learning contribution of this work is to show that synthetic fractal images can serve as a useful resource for self-supervised representation learning when domain-specific data are limited.
>
> Although we evaluate only two downstream tasks—and supervised learning performs competitively for stellar-mass prediction—the results suggest that fractal pretraining can achieve accuracy comparable to supervised learning while providing representations that may generalize more broadly. We view this as an initial indication that domain-inspired synthetic data could support flexible SSL pipelines for fluid dynamics simulations, and we hope it encourages further exploration in this direction.
>
> # 4 Connection between $R^2 = 0.81$ and the scientific goal of stellar-mass inference
>
> Our broader objective is to develop an emulator that estimates stellar masses from their surrounding environments. In observational settings, stellar masses are inferred indirectly from various measurable properties, and the resulting accuracy can vary substantially. In favorable cases, uncertainties can be on the order of tens of percent, while in more challenging conditions, they may increase to roughly a factor of two.
> The performance we report—$R^2 = 0.81$—shows that the model captures about 80% of the total variance in stellar masses. This level of accuracy is lower than what is achievable under the best observational conditions, but is comparable to the uncertainty levels encountered in less optimal scenarios.
>
> Consequently, while the current model already provides a meaningful indication of large-scale correlations between local environments and stellar masses, it is not yet precise enough to function as a practical scientific emulator. Further improvements—such as incorporating additional input features, refining the model architecture, and performing task-specific fine-tuning—will be required to reduce errors. With these enhancements, the model could potentially reach accuracy comparable to high-quality observational estimates.

---

### Meta-Review · Area_Chair_SpQs · 2026-01-06

**Summary:**

The authors were unable to submit all rebuttals by the deadline.

**Reviewer Concerns:**

Not available as some of the rebuttals are still missing.

**Reviewer Scores:**

N/A

---

### Decision · Program_Chairs · 2026-01-26

Reject